# Optical Transmission in Single-Layer Brain Tissues under Different Optical Source Types: Modelling and Simulation

**DOI:** 10.3390/bioengineering11090916

**Published:** 2024-09-13

**Authors:** Xi Yang, Chengpeng Chai, Yun-Hsuan Chen, Mohamad Sawan

**Affiliations:** 1CenBRAIN Neurotech Center of Excellence, School of Engineering, Westlake University, 600 Dunyu Road, Xihu District, Hangzhou 310030, China; yangxi@westlake.edu.cn (X.Y.); chaichengpeng@westlake.edu.cn (C.C.); 2Institute of Advanced Technology, Westlake Institute for Advanced Study, 18 Shilongshan Street, Xihu District, Hangzhou 310024, China

**Keywords:** photoacoustic imaging, optical imaging, brain imaging, optical simulation, Monte Carlo simulation, single-layer brain tissue

## Abstract

The human brain is a complex organ controlling daily activity. Present technique models have mostly focused on multi-layer brain tissues, which lack understanding of the propagation characteristics of various single brain tissues. To better understand the influence of different optical source types on individual brain tissues, we constructed single-layer brain models and simulated optical propagation using the Monte Carlo method. Based on the optical simulation results, sixteen optical source types had different optical energy distributions, and the distribution in cerebrospinal fluid had obvious characteristics. Five brain tissues (scalp, skull, cerebrospinal fluid, gray matter, and blood vessel) had the same set of the first three optical source types with maximum depth, while white matter had a different set of the first three optical source types with maximum depth. Each brain tissue had different optical source types with the maximum and minimum full width at half maximum. The study on single-layer brain tissues under different optical source types lays the foundation for constructing complex brain models with multiple tissue layers. It provides a theoretical reference for optimizing the selection of optical source devices for brain imaging.

## 1. Introduction

The human brain is a complex organ [1], which controls daily activities such as action, emotion, and thinking. With the exploration of brain function and diseases, detection techniques have been developed based on different principles, such as X-ray computed tomography (X-ray CT), magnetic resonance imaging (MRI), electroencephalogram (EEG), functional near-infrared spectroscopy (fNIRS), optical imaging, ultrasound imaging, and photoacoustic imaging [2]. To achieve high resolution and deep imaging depth, photoacoustic imaging has been developed for brain imaging with the advantages of optical specificity and acoustic penetration. However, with the complex structure of the human brain, especially the skull layer, it is still challenging for human brain imaging [3].

Due to the attenuation and distortion of the signals in the human brain, related research works were conducted to improve the imaging performance of PAI for the human brain. The research work includes the exploration of PA signal generation and propagation [4], as well as the design and optimization of the parameters and structure for the system [5] in the aspects of optics [6], acoustics [7,8,9], and signal processing [10]. According to the aforementioned research, modeling and simulation are crucial for understanding the transmission of optical and ultrasound signals in brain tissues, especially the human skull.

In the research of the optical aspect, the researchers contributed a lot of work to understand and improve the performance of brain imaging. To analyze the influence of the skull on optical transmission, Song et al. researched the optical simulation work of a five-layer brain model based on the Monte Carlo method using the MCML simulation program [11]. Moreover, Song et al. also researched the mouse brain model using the Monte Carlo simulation method [12]. To improve the illumination of the PAI in an infant’s brain, Mahmoodkalayeh et al. proposed a method to achieve uniform fluence within the region of interest (ROI) based on the Monte Carlo simulation [13]. In this work, the optical sources with different positions, diameters, and numerical apertures were researched to find the optical source configuration for the best fluence distribution. Previously, we researched the influence of different optical source types on multi-layered brain tissues based on the Monte Carlo method [14]. This research can determine the performance of the optical source types in the specific brain model. However, the brain models of different patients are different. Meanwhile, the brain tissues with uncertainty vary from tissue to tissue. Therefore, exploring the influence of optical source types in each brain tissue is important to better understand the complex brain model.

In this paper, we constructed simulation models for single-layer brain tissues to explore the optical propagation and diffusion under different types of incident light sources. Simulation models for single-layer brain tissues were built based on the Monte Carlo method, which considered the optical properties of six brain tissues including the scalp, skull, cerebrospinal fluid (CSF), gray matter, white matter, and blood vessels. By applying quantitative analysis methods, the propagation depth and diffusion characteristics of different light source types in each tissue were evaluated to determine the optimal light source type for each tissue.

## 2. Materials and Methods

### 2.1. Simplified Brain Model

The human brain is complex with multi-layer tissues. The brain cortex is protected mainly by the scalp, skull, and CSF. The brain cortex includes gray matter and white matter. Meanwhile, blood vessels surround the brain cortex. Therefore, we built six single-layer brain models representing the six kinds of brain tissues: the scalp, skull, CSF, gray matter, white matter, and blood vessels. In this study, we employed a single-layer brain model to simplify the analysis process. This model assumes that the brain tissue is homogeneous, focusing on the interactions between optical source types with this uniform medium. The brain model was a 14 × 14 × 14 mm^3^ volume with a unit size of 0.1 mm to ensure enough space for optical photon propagation (Figure 1).

For optical simulation, the optical properties of six brain tissues under an incident light at 800 nm were listed in Table 1 [15]. The 800 nm wavelength lies within the near-infrared spectrum, where light absorption by water and hemoglobin is relatively low. This results in reduced scattering and absorption, allowing for deeper tissue penetration, which is particularly beneficial for neurological simulations. Furthermore, this wavelength is commonly used in photoacoustic imaging, making it a relevant choice for comparison and consistency with the existing literature and experimental setups [16].

### 2.2. Optical Simulation Method

The optical simulation was performed in a 3D space by an open-source MATLAB toolbox named ‘mcxlab’ based on the Monte Carlo method [17]. The software of MATLAB R2019b was used in the research. Meanwhile, the version of the MATLAB toolbox was ‘mcxlab-win-x86_64-nightlybuild’. The photon number used for simulation was set at 10^8^. The simulation time was set at 5 × 10^−8^ s for both ending time and time-gate width. The volume of the simulation model was 140 × 140 × 140. The grid size was set as 0.1 mm. Based on the simulation, the optical fluence was acquired for analysis. As known, optical absorption is related to optical fluence as shown in Equation (1) [18],
(1)$$A=\mu_{a}F$$
where *A* represents the optical absorption, *F* represents the optical fluence, and *μ_a_* depicts the absorption coefficient.

### 2.3. Various Types of Optical Sources

In this article, we used sixteen optical source types which were mentioned for the multi-layer brain model in reference [14]. The types of the sixteen optical sources were named pencil, isotropic, cone, arcsine, line, slit, collimated Gaussian, angular Gaussian, hyperboloid Gaussian, planar, disk, ring, pencil array, spatial frequency Fourier, 1D Fourier, and 2D Fourier sources. The difference between those optical source types is related to the shape, size, distribution, focus, diffusion, and others according to the definitions. The settings of each optical source were conducted following reference [14] in detail, which were used to research the difference in optical propagation inside the single-layer brain models in this research. The optical sources launched at the position (*x* = 7 mm, *y* = 7 mm, *z* = 0 mm) and propagated vertically toward the depth-increasing direction.

### 2.4. Data Analysis

#### 2.4.1. Data Preprocessing

After optical simulation, the optical fluence was acquired for analysis. The absorption coefficients in our simulation models are homogeneous; there is a linear relationship between optical absorption and optical fluence in each model. Therefore, we only analyzed the optical fluence after normalization, which was defined as the optical energy in this research.

The optical energy was normalized from 0 to 100 based on Equation (2),
(2)$${E_{N}}_{x,y,z}=\frac{E_{x,y,z}-E_{\min}}{E_{\max}-E_{\min}}\times 100$$
where E_N*x*,*y*,*z*_ represents the normalized optical energy at the position (*x*, *y*, *z*), E*_x_*_,*y*,*z*_ named the optical energy at the position (*x*, *y*, *z*), E_max_ donates the maximum value of the optical energy, and E_min_ means the minimum value of the optical energy.

#### 2.4.2. Propagating Depth Evaluation

The optical propagating depth was evaluated based on the optical energy in the Z-axis direction at the plane of *x* = 7 mm and *y* = 7 mm. The change of the optical energy can be observed in the depth direction. The results of the six brain tissues under the specific illumination were drawn in one subfigure. The results of sixteen optical source types were compared.

Moreover, the maximum depth of each optical source at three optical energy levels was extracted. The three levels were 1%, 0.1%, and 0.01% of the maximum optical energy. Meanwhile, the corresponding optical source types with the maximum depth were marked.

#### 2.4.3. Optical Field Width Evaluation

In this part, the optical field width was evaluated to describe the optical diffusion at a certain depth. The full width at half maximum (FWHM) was calculated based on the width at half of the maximum height of the optical energy. As depicted in Figure 2, the optical energy from the horizontal X-axis and Y-axis directions was extracted at a specific depth. The peak energy in the current direction was identified, and the corresponding full width was determined from the points on two sides of this peak where the energy dropped to half, above the minimum energy level. Furthermore, the FWHM values and the types of sources corresponding to a depth of 12 mm were systematically compared, paralleling the methodology used in reference [14].

## 3. Results and Discussion

### 3.1. Optical Distribution in Different Planes

The energy distribution of optical fluence and absorption looks the same due to the linear relationship of each other. Therefore, only the optical energy distribution after normalization with log operation by optical fluence is discussed in this part.

As a typical example of optical source types, we chose the pencil beam because of its highly focused and directional characteristics, which allow us to precisely study light propagation in brain tissue. Additionally, the pencil beam is commonly used for baseline analyses, aiding in fundamental research and comparative experiments. The results of other optical sources were summarized in the Appendix A. Figure 3 shows the optical energy distribution in six kinds of single-layer brain tissue models under the illumination of a pencil beam. It can be seen from the figure that the pencil beam was very narrow and focused in the XY plane, which had a very small divergence angle and strong characteristic of straight-line propagation in XZ and YZ planes. Due to the low absorption and scattering coefficients of CSF, the high focusing characteristics of the beam can be seen in Figure 3g–i, whose light spot was very small.

Here, we compared the optical energy distribution in different brain tissues. In Figure 3, it can be observed that the optical propagation distance in the scalp and skull became shorter than the CSF, and the optical energy attenuated rapidly because of the higher absorption and scattering coefficients of the scalp and skull compared with the CSF. The white matter had a stronger absorption coefficient than the scalp and skull.

Meanwhile, the optical attenuation and diffusion were analyzed. It was found that the optical attenuation and diffusion range in Figure 3m–o were larger than the scalp and skull in Figure 3a–f. Rather than white matter, the gray matter had a larger absorption coefficient, and had rapid optical attenuation with a smaller propagation depth. Figure 3g–p shows the optical energy in blood tissue.

### 3.2. Optical Propagation in Vertical Direction

Figure 4 shows the optical energy of the six brain tissues in the Z direction (*x* = 7 mm, and *y* = 7 mm) under the illumination of different optical source types. The simulation results in the 3D volume were normalized in the range of 0 to 100. In each subfigure, it can be found that the six brain tissues under the same optical source types had the same tendency while the amplitude of the tendency was different in the different optical source types. Based on Figure 4b–e,h, the amplitude of the change tendency looked not obvious, which shows strong overlapping under the illumination of isotropic, cone, arcsine, line, and angular Gaussian sources. Figure 4a,f shows that the overlapping region was after 4 mm under the illumination of the pencil and slit sources.

Meanwhile, Figure 4g,i–p indicates an obvious difference at the region of 0–6 mm under the illumination of the collimated Gaussian, hyperboloid Gaussian, planar, disk, ring, pencil array, spatial frequency Fourier, 1D Fourier, and 2D Fourier sources. Furthermore, according to Figure 4, it can be found that the difference in the tendency mainly happened in the shallow region within 8 mm, especially within 4 mm. When the depth went deeper, the difference in the results was not too obvious. Considering the difference between the sixteen optical source types, it can be seen that the tendency of the ring source increased first and then decreased, which is the same as the pencil array. In addition, the beginning of the ring, pencil array, spatial frequency Fourier, and 2D Fourier sources were not the maximum values in the volume.

Figure 5 shows the maximum depth of six brain tissues at the specific optical energy levels of 1%, 0.1%, and 0.01% under the different optical source types in the Z direction with *x* = 7 mm and *y* = 7 mm. Based on the results, it can be found that the maximum depth of five tissues except for the white matter in the various levels had the same tendency. In addition, the optical source types with the maximum depth were the same in the five tissues and the optical energy level. The first three optical source types with maximum depth in the five tissues were collimated Gaussian, planar, and disk sources. In comparison, the first three optical source types with maximum depth in the gray matter were collimated Gaussian, hyperboloid Gaussian, and planar sources. Moreover, we found that the spatial frequency Fourier source did not have a depth at the level of 1%, which means that the optical energy of the spatial frequency was smaller than the energy level of 1% in the Z direction with *x* = 7 mm and *y* = 7 mm. Furthermore, more than three optical source types had the maximum depth at the optical energy of 0.01% in the skull and CSF tissues.

Here, it can be known that the light is focused and gradually diffused in the deeper region owing to the highest absorption coefficient and higher scattering coefficient.

### 3.3. Optical Distribution in Horizontal Directions

Figure 6 shows the results of the FWHM of the optical energy at the X and Y axes for six brain tissues. Based on the results, the distribution of the FWHM of the energy at the X and Y axes can be observed. In the overview, the FWHM became larger (from less than 1 mm to more than 9 mm) with the depth increasing (from 0.1 mm to 14 mm). Obviously, the FWHM of hyperboloid Gaussian at the X and Y axes was larger than the other sources, especially in the CSF and blood vessel tissues. And isotropic source was the second largest in both the X and Y axes, while the line source was only obviously larger in the Y axis.

Figure 7 illustrates the FWHM of optical energy in the X and Y axes at a depth of 12 mm. According to the figure, it was found that the FWHMs varied from the optical source types and tissues. The maximum FWHM was 9.1 mm in the scalp tissue under the hyperboloid Gaussian source. The minimum FWHM was 6.8 mm in the blood vessel tissue under the spatial frequency Fourier. The largest distance of the FWHM in each tissue was found in the CSF tissue at 1.9 mm.

### 3.4. Optical Source Types under Different Conditions

Based on the linear relationship between optical fluence and absorption, it can be known that those two parameters have the same tendency only with the difference in the value. So those two parameters were represented by the optical energy. Then we compared the results in the six single-layered brain tissues under the illumination of sixteen optical source types. Furthermore, based on Figure 5 and Figure 7, we have summarized the optical source types and the corresponding value in the first three or last three in the penetration depth in Table 2 and field width in Table 3. In Table 2, if there were several same maximum depths, the optical source types with high values in most conditions were selected.

Considering the penetration depth according to Table 2, the first three optical source types with maximum depths at the optical energy levels of 1%, 0.1%, and 0.01% were collimated Gaussian, planar, and disk in the five single-layer brain tissues including the scalp, skull, CSF, gray matter, and blood vessel, while the first three optical source types with maximum depths at the optical energy of 1%, 0.1%, and 0.01% in the white matter were collimated Gaussian, hyperboloid Gaussian, and planar sources.

For field width to evaluate the resolution, the results were analyzed in the FWHM of the optical energy in the X and Y axes in Table 3. Different single-layered brain tissues had the different first three maximum and minimum FWHM of the optical energy and corresponding optical source types. The first three optical source types with the maximum FWHM in the X axis in the scalp were hyperboloid Gaussian, planar, and 2D Fourier sources, while the first three optical source types in the Y axis were pencil, line, and hyperboloid Gaussian sources. The first three optical source types with the minimum FWHM in the X axis in the scalp were disk, collimated Gaussian, and slit sources, while the first three optical source types in the Y axis were disk, arcsine, and cone sources.

The first three optical source types with the maximum FWHM in the X axis in the skull were cone, pencil, and 2D Fourier sources, while the first three optical source types in the Y axis were isotropic, arcsine, and planar sources. The first three optical source types with the minimum FWHM in the X axis in the skull were 2D Fourier, pencil, and angular Gaussian sources, while the first three optical source types in the Y axis were slit, ring, and pencil sources.

The first three optical source types with the maximum FWHM in the X axis in the CSF were hyperboloid Gaussian, arcsine, and cone sources, while the first three optical source types in the Y axis were hyperboloid Gaussian, isotropic, and cone sources. The first three optical source types with the minimum FWHM in the X axis in the CSF were spatial frequency, 1D Fourier, and 2D Fourier sources, while the first three optical source types in the Y axis were spatial frequency Fourier, planar, and arcsine sources.

The first three optical source types with the maximum FWHM in the X axis in the gray matter were hyperboloid Gaussian, arcsine, and 1D Fourier sources, while the first three optical source types in the Y axis were line, angular Gaussian, and 2D Fourier sources. The first three optical source types with the minimum FWHM in the X axis in the gray matter were slit, collimated Gaussian, and ring sources, while the first three optical source types in the Y axis were ring, slit, and disk sources.

The first three optical source types with the maximum FWHM in the X axis in the white matter were hyperboloid Gaussian, ring, and slit sources, while the first three optical source types in the Y axis were line, arcsine, and disk sources. The first three optical source types with the minimum FWHM in the X axis in the white matter were angular Gaussian, isotropic, and cone sources, while the first three optical source types in the Y axis were ring, spatial frequency, and angular Gaussian sources.

The first three optical source types with the maximum FWHM in the X axis in the blood vessel were hyperboloid Gaussian, arcsine, and angular Gaussian sources, while the first three optical source types in the Y axis were collimated Gaussian, arcsine, and isotropic sources. The first three optical source types with the minimum FWHM in the X axis in the blood vessel were collimated Gaussian, disk, and isotropic sources, while the first three optical source types in the Y axis were spatial frequency Fourier, ring, and pencil array sources.

From the results, the hyperboloid Gaussian source was the first one with the maximum FWHM in the X axis of all tissues. Arcsine source was the second maximum FWHM in the X axis in the three tissues (CSF, gray matter, and blood vessel).

### 3.5. Discussion with Multi-Layer Brain Model

The human brain is a complex organ with multi-layer brain tissues. Multi-layer brain models have been used for simulation works with accurate results, which are valuable in clinical applications and disease diagnosis. Previously, we conducted the optical simulation of the multi-layer brain models under the illumination of different optical source types [14]. However, the complexity of the multi-layer structure makes it difficult to capture the signal propagation in the specific tissues to understand the tissue diversity. In this paper, we researched the optical transmission of the single-layer brain models under the illumination of the different optical source types, which reduced the complexity introduced by multi-layer structures. Therefore, we compared those two studies to understand the characteristics caused by tissue uncertainty.

From the comparison, it was found that the optical energy in the multi-layer brain model changed layer by layer, especially the optical absorption. In contrast, the optical energy in the single-layer brain model changed steadily. The reason is that the single-layer brain model has high homogeneity. However, the multi-layer brain model was influenced by several brain tissues with different optical properties. Thus, the multi-layer brain model was obviously a stage change between the boundary of two layers.

Furthermore, it was obvious that the optical energy under the specific optical source types had different levels due to the difference in the tissues in the single-layer brain models. In the multi-layer brain model, it was clear that the difference in the surface region was due to the optical source types, the difference in the deep region depended on the levels of the optical energy.

According to the research on the optical transmission of the single-layer and multi-layer brain models, the methods can be followed for the other specific brain model research. Meanwhile, the selection of the optical source devices can be further improved for brain imaging.

## 4. Conclusions

With the lack of understanding of the propagation characteristics of various single brain tissues under different optical source types, we constructed single-layer brain tissue models and optical simulations using the Monte Carlo method. Six brain tissues were analyzed including scalp, skull, CSF, gray matter, white matter, and blood vessel. Meanwhile, sixteen optical source types were individually illuminated in the six brain tissues. Based on the optical simulation results, sixteen optical source types had different optical energy distributions, and the distribution in CSF had obvious characteristics. Five brain tissues (scalp, skull, CSF, gray matter, and blood vessel) had the same set of the first three optical source types with the maximum depth, while white matter had a different set of the first three optical source types with maximum depth. Each brain tissue had different optical source types with the maximum and minimum FWHM. Hyperboloid Gaussian source had the maximum FWHM at the X-axis direction of all tissues. The study on single-layer brain tissues under different optical source types lays the foundation for constructing complex brain models with multiple tissue layers. It provides a theoretical reference for optimizing the selection of optical source devices for brain imaging, which is of great significance for improving human brain photoacoustic imaging and other optical-related techniques. In the future, the other characteristics of the optical sources in optical intensity and distribution can also be researched to improve the illumination of optical sources under different situations.

## Figures and Tables

**Figure 1 bioengineering-11-00916-f001:**
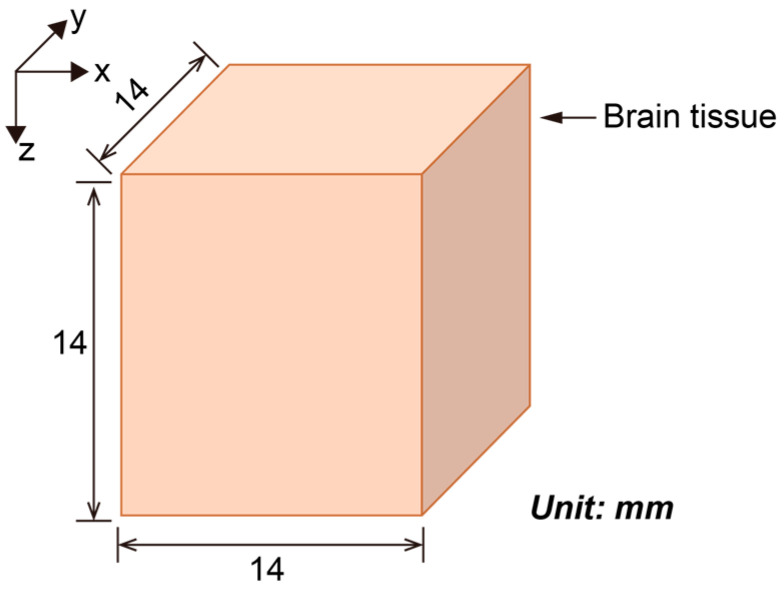
Structure of a simplified brain model of an individual layer of tissue.

**Figure 2 bioengineering-11-00916-f002:**
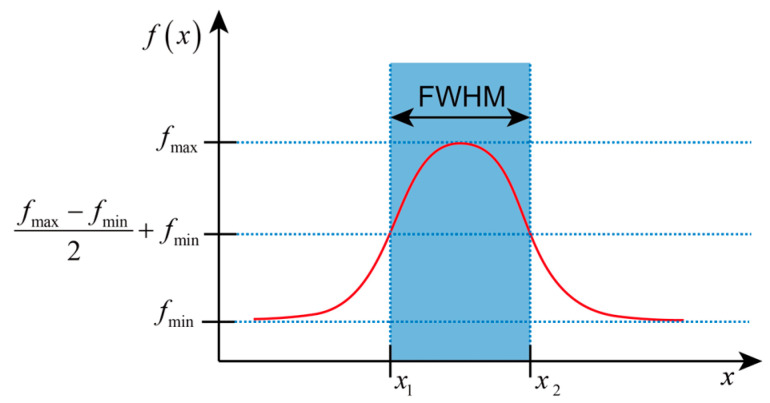
Optical field width evaluation based on the full width at half maximum (FWHM). The red line represents the energy distribution function *f*(*x*) of position *x*, while the blue shaded area represents the FWHM region of that distribution.

**Figure 3 bioengineering-11-00916-f003:**
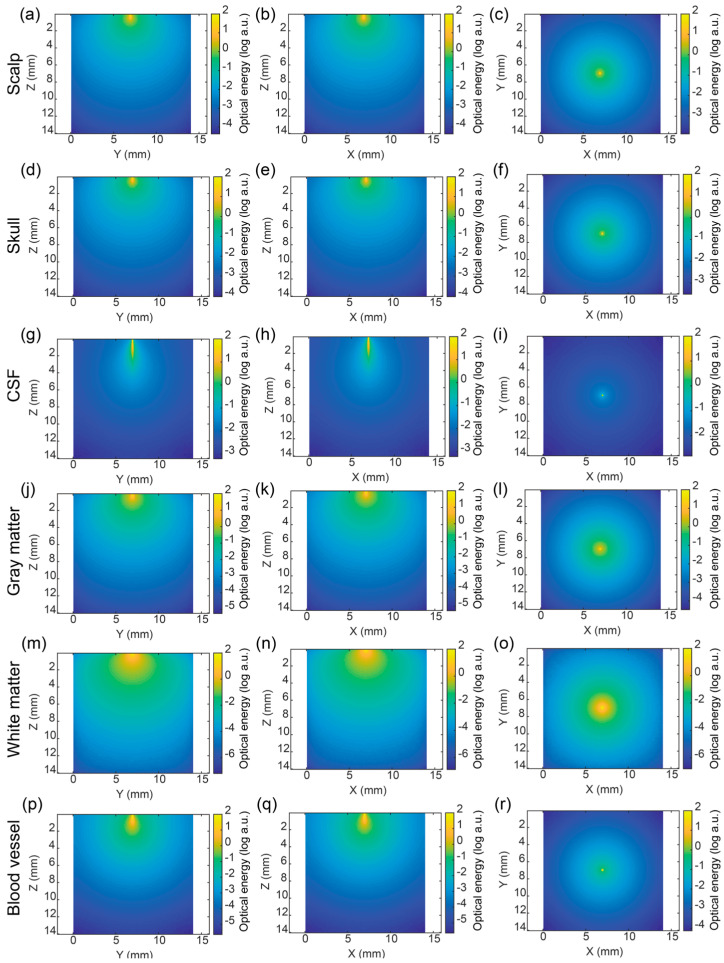
Optical energy distribution of single-layer brain tissues under the illumination of a pencil beam. (**a**–**c**) Scalp, (**d**–**f**) skull, (**g**–**i**) CSF, (**j**–**l**) gray matter, (**m**–**o**) white matter, and (**p**–**r**) blood vessel; (**a**,**d**,**g**,**j**,**m**) on the YZ plane (*x* = 7 mm), (**b**,**e**,**h**,**k**,**n**) on the XZ plane (*y* = 7 mm), (**c**,**f**,**i**,**l**,**r**) the XY plane (*z* = 0.1 mm), respectively.

**Figure 4 bioengineering-11-00916-f004:**
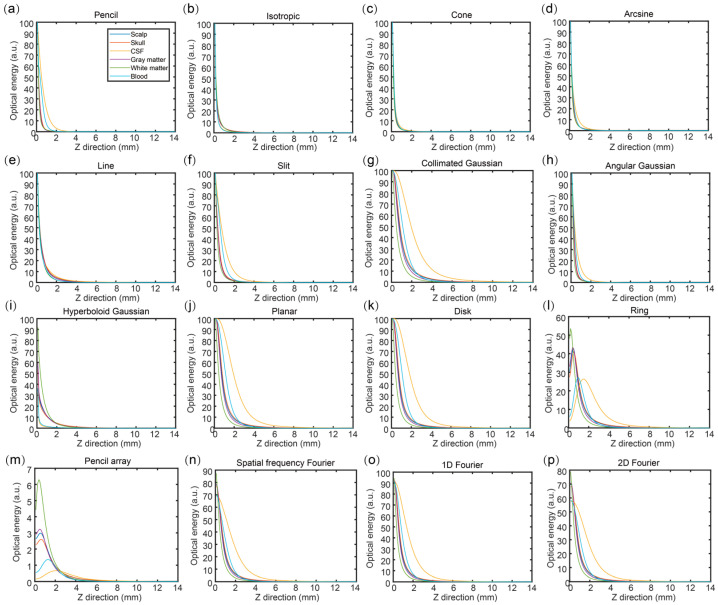
Optical energy of six brain tissues under the illumination of different optical source types in the Z direction with *x* = 7 mm and *y* = 7 mm. (**a**) Pencil; (**b**) Isotropic; (**c**) Cone; (**d**) Arcsine; (**e**) Line; (**f**) Slit; (**g**) Collimated Gaussian; (**h**) Angualr Gaussian; (**i**) Hyperiboloid Gaussian; (**j**) Planar; (**k**) Disk; (**l**) Ring; (**m**) Pencil array; (**n**) Spatial frequency Fourier; (**o**) 1D Fourier; (**p**) 2D Fourier.

**Figure 5 bioengineering-11-00916-f005:**
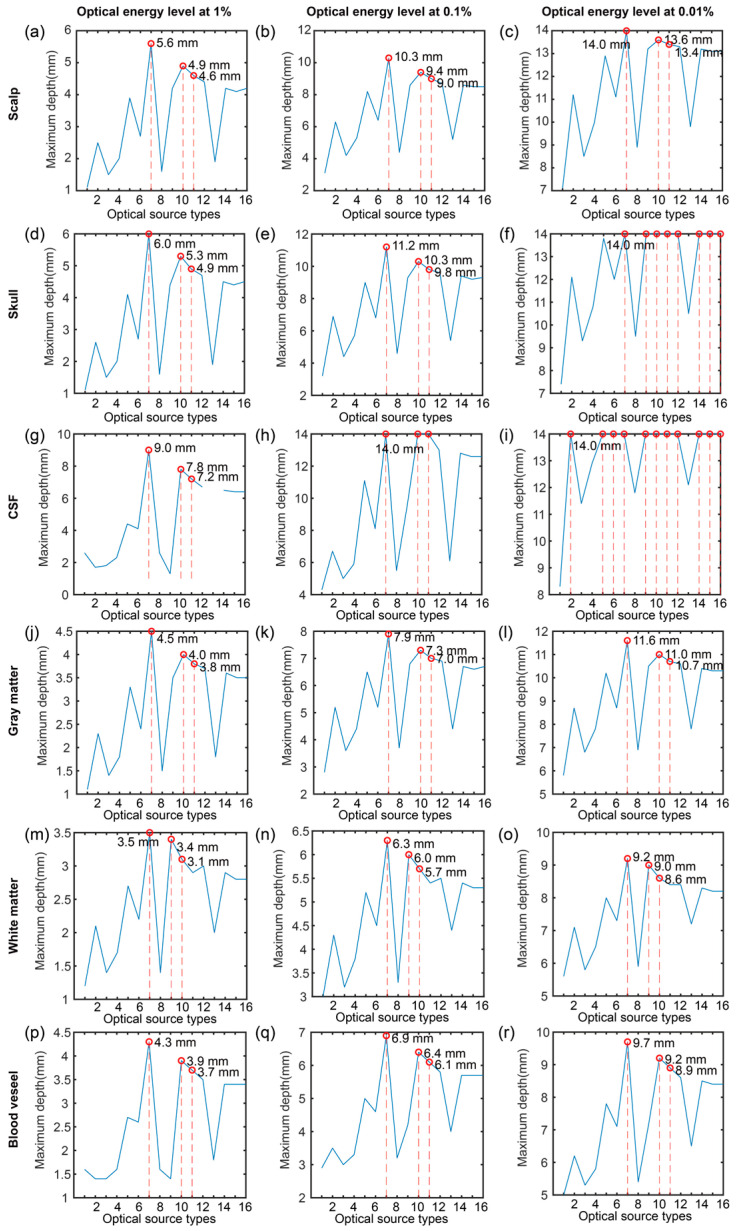
Maximum depth of six brain tissues at specific optical energy levels of 1%, 0.1%, and 0.01% under the different optical source types in the Z direction with *x* = 7 mm and *y* = 7 mm. (**a**–**c**) Scalp; (**d**–**e**) Skull; (**g**–**i**) CSF; (**j**–**l**) Gray matter; (**m**–**o**) White matter; (**p**–**r**) Blood vessel. 1—pencil, 2—isotropic, 3—cone, 4—arcsine, 5—line, 6—slit, 7—collimated Gaussian, 8—angular Gaussian, 9—hyperboloid Gaussian, 10—planar, 11—disk, 12—ring, 13—pencil array, 14—spatial frequency Fourier, 15—1D Fourier, and 16—2D Fourier. The red color marks the maximum value.

**Figure 6 bioengineering-11-00916-f006:**
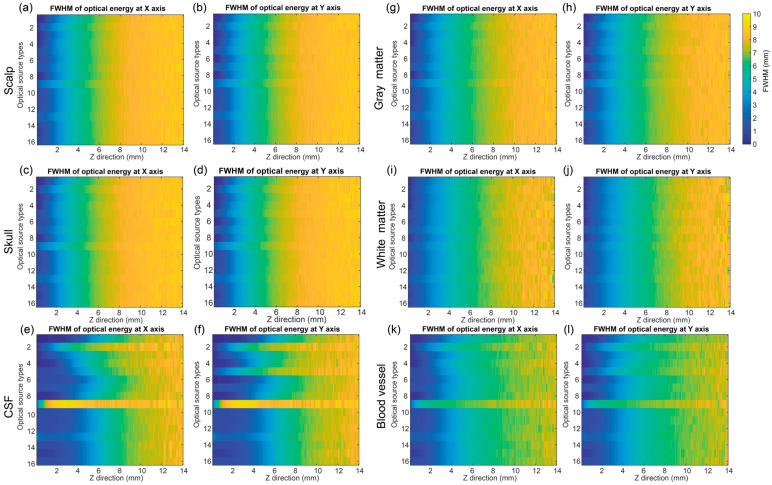
The FWHM of the optical energy at the X and Y axes for six brain tissues. (**a**,**b**) Scalp, (**c**,**d**) Skull, (**e**,**f**) CSF, (**g**,**h**) Gray matter, (**i**,**j**) White matter, and (**k**,**l**) Blood vessel. 1—pencil, 2—isotropic, 3—cone, 4—arcsine, 5—line, 6—slit, 7—collimated Gaussian, 8—angular Gaussian, 9—hyperboloid Gaussian, 10—planar, 11—disk, 12—ring, 13—pencil array, 14—spatial frequency Fourier, 15—1D Fourier, and 16—2D Fourier.

**Figure 7 bioengineering-11-00916-f007:**
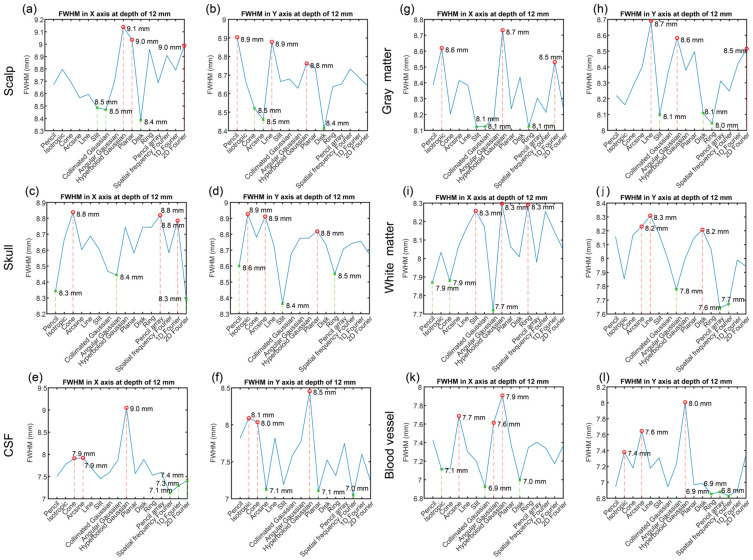
The FWHM of the optical energy in the X and Y axes at the *z* of 12 mm for six brain tissues. (**a**,**b**) Scalp, (**c**,**d**) Skull, (**e**,**f**) CSF, (**g**,**h**) Gray matter, (**i**,**j**) White matter, and (**k**,**l**) Blood vessel. Red color marks the maximum value, green color marks the minimum value.

**Table 1 bioengineering-11-00916-t001:** Optical properties of the brain tissues at the optical wavelength of 800 nm.

Brain Tissues	Absorption Coefficient, *μ_a_* (1/mm)	Scattering Coefficient, *μ_s_* (1/mm)	Anisotropy Factor, *g*	Refractive Index, *n*
Scalp	0.018	19.0	0.9	1.37
Skull	0.016	16.0	0.9	1.43
Cerebrospinal fluid	0.004	2.4	0.9	1.33
Gray matter	0.036	22.0	0.9	1.37
White matter	0.014	91.0	0.9	1.37
Blood vessel	0.223	50.0	0.99	1.4

**Table 2 bioengineering-11-00916-t002:** The first three optical source types with the deepest penetration depth in each brain layer.

Tissue	Optical Energy of 1%	Optical Energy of 0.1%	Optical Energy of 0.01%
Scalp	Collimated Gaussian (5.6 mm)	Collimated Gaussian (10.3 mm)	Collimated Gaussian (14.0 mm)
	Planar (4.9 mm)	Planar (9.4 mm)	Planar (13.6 mm)
	Disk (4.6 mm)	Disk (9.0 mm)	Disk (13.4 mm)
Skull	Collimated Gaussian (6.0 mm)	Collimated Gaussian (11.2 mm)	Collimated Gaussian (14.0 mm)
	Planar (5.3 mm)	Planar (10.3 mm)	Planar (14.0 mm)
	Disk (4.9 mm)	Disk (9.8 mm)	Disk (14.0 mm)
CSF	Collimated Gaussian (9.0 mm)	Collimated Gaussian (11.2 mm)	Collimated Gaussian (14.0 mm)
	Planar (7.8 mm)	Planar (10.3 mm)	Planar (14.0 mm)
	Disk (7.2 mm)	Disk (9.8 mm)	Disk (14.0 mm)
Gray matter	Collimated Gaussian (4.5 mm)	Collimated Gaussian (7.9 mm)	Collimated Gaussian (11.6 mm)
	Planar (4.0 mm)	Planar (7.3 mm)	Planar (11.0 mm)
	Disk (3.8 mm)	Disk (7.0 mm)	Disk (10.7 mm)
White matter	Collimated Gaussian (3.5 mm)	Collimated Gaussian (6.3 mm)	Collimated Gaussian (9.2 mm)
	Hyperboloid Gaussian (3.4 mm)	Hyperboloid Gaussian (6.0 mm)	Hyperboloid Gaussian (9.0 mm)
	Planar (3.1 mm)	Planar (5.7 mm)	Planar (8.6 mm)
Blood vessel	Collimated Gaussian (4.3 mm)	Collimated Gaussian (6.9 mm)	Collimated Gaussian (9.7 mm)
	Planar (3.9 mm)	Planar (6.4 mm)	Planar (9.2 mm)
	Disk (3.7 mm)	Disk (6.1 mm)	Disk (8.9 mm)

**Table 3 bioengineering-11-00916-t003:** The first three maximum and minimum field widths and corresponding optical source types in each brain layer.

Tissue	Optical Source Types with Maximum Value		Optical Source Types with Minimum Value	
	FWHM of Energy in the X Axis	FWHM of Energy in the Y Axis	FWHM of Energy in the X Axis	FWHM of Energy in the Y Axis
Scalp	Hyperboloid Gaussian (9.1 mm)	Pencil (8.9 mm)	Disk (8.4 mm)	Disk (8.4 mm)
	Planar (9.0 mm)	Line (8.9 mm)	Collimated Gaussian (8.5 mm)	Arcsine (8.5 mm)
	2D Fourier (9.0 mm)	Hyperboloid Gaussian (8.8 mm)	Slit (8.5 mm)	Cone (8.5 mm)
Skull	Cone (8.8 mm)	Isotropic (8.8 mm)	2D Fourier (8.3 mm)	Slit (8.4 mm)
	Pencil (8.8 mm)	Arcsine (8.9 mm)	Pencil (8.3 mm)	Ring (8.5 mm)
	2D Fourier (8.8 mm)	Planar (8.8 mm)	Angular Gaussian (8.4 mm)	Pencil (8.6 mm)
CSF	Hyperboloid Gaussian (9.0 mm)	Hyperboloid Gaussian (8.5 mm)	Spatial frequency Fourier (7.1 mm)	Spatial frequency Fourier (7.0 mm)
	Arcsine (7.9 mm)	Isotropic (8.1 mm)	1D Fourier (7.3 mm)	Planar (7.1 mm)
	Cone (7.9 mm)	Cone (8.0 mm)	2D Fourier (7.4 mm)	Arcsine (7.1 mm)
Gray matter	Hyperboloid Gaussian (8.7 mm)	Line (8.7 mm)	Slit (8.1 mm)	Ring (8.0 mm)
	Arcsine (8.6 mm)	Angular Gaussian (8.6 mm)	Collimated Gaussian (8.1 mm)	Slit (8.1 mm)
	1D Fourier (8.5 mm)	2D Fourier (8.5 mm)	Ring (8.1 mm)	Disk (8.1 mm)
White matter	Hyperboloid Gaussian (8.3 mm)	Line (8.3 mm)	Angular Gaussian (7.7 mm)	Ring (7.6 mm)
	Ring (8.3 mm)	Arcsine (8.2 mm)	Isotropic (7.9 mm)	Spatial frequency Fourier (7.7 mm)
	Slit (8.3 mm)	Disk (8.2 mm)	Cone (7.9 mm)	Angular Gaussian (7.8 mm)
Blood vessel	Hyperboloid Gaussian (7.9 mm)	Collimated Gaussian (8.0 mm)	Collimated Gaussian (6.9 mm)	Spatial frequency Fourier (6.8 mm)
	Arcsine (7.7 mm)	Arcsine (7.6 mm)	Disk (7.0 mm)	Ring (6.9 mm)
	Angular Gaussian (7.6 mm)	Isotropic (7.4 mm)	Isotropic (7.1 mm)	Pencil array (6.9 mm)

## Data Availability

The datasets generated for this study are available on request to the corresponding authors.

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
