# Peer review of "Optical Transmission in Single-Layer Brain Tissues under Different Optical Source Types: Modelling and Simulation"

_bioengineering, 2024, doi:10.3390/bioengineering11090916_

Round 1
Reviewer 1 Report
Comments and Suggestions for Authors
his manuscript solves the problem of the absorption and scattering characteristics of light sources in different layers of brain tissue, which is an interesting topic I have been concerned about. Besides, it lists in detail the penetration depth and optical diffusion of different types of light sources in this brain tissue, which is highly innovative and instructive, and I suggest publishing this paper. Moreover, there are some specific issues that need to be improved.
Lines 41-44, the sentence could be improved: “The simulation method is very important for the above research works, especially for understanding the transmission of optical and ultrasound signals through the human skull based on the simulation methods.”
Lines 67-68, the sentence could be improved from “determining the optimal light source type for each tissue.” to “ to determine the optimal light source type for each tissue.”
The Italic and normal fonts of “x, y, z” should be consistent in Equation 2 and Line 111.
Lines 115-116, “x”, “y”, and “z” for axis direction should be consistent in capital and small letters in the manuscript.
Line 217, “Figure 3” should be “Figure 2”.
In conclusion, perspectives could be added for further works, like other characteristics of the optical sources in optical intensity, and distribution, etc.
Reviewer 2 Report
Comments and Suggestions for Authors
This manuscript describes the light transmission characteristics of 16 light sources in the different brain tissues. It has a very good guiding significance for the development of relevant medical devices in the later stage. I agree to publish this manuscript. Still, before publishing, there are some questions to answer.
1. Different tissues and optical properties obviously make different results. Why is it necessary to do the simulation of single-layer brain models not only the multi-layer brain models?
2. What’s the difference between single-layer brain models and multi-layer brain models?
3. What is the difference between 16 optical source types? Could you give more information on this manuscript?
4. Why was the optical wavelength of 800 nm selected for the simulation?
5. Why was the pencil beam shown in Section 3.1? How about the results of other optical sources?
Comments on the Quality of English LanguageMinor editing of English language required.
